# Full field assessment of wind turbine near wake deviation in relation to yaw misalignment

Juan-José Trujillo[1], Janna K. Seifert[1], Ines Würth[2], David Schlipf[2], and Martin Kühn[1]

[1]ForWind - University of Oldenburg, Institute of Physics, Ammerländer-Heerstr. 136, 26129 Oldenburg, Germany
[2]Stuttgart Wind Energy at University of Stuttgart, Institute of Aircraft Design, Allmandring 5B, 70569 Stuttgart, Germany

*Correspondence to:* J.J. Trujillo (juan.jose.trujillo@uni-oldenburg.de)

**Abstract.** Presently there is a lack of data revealing the behaviour of the path followed by the near wake of full scale wind turbines and its dependence on yaw misalignment. Here we present an experimental analysis of the horizontal wake deviation of a 5 MW offshore wind turbine between 0.6 and 1.4 diameters downstream. The wake field has been scanned with a short range lidar and the wake path has been reconstructed by means of two-dimensional Gaussian tracking. We analysed the measurements
for rotor yaw misalignments arising in normal operation and during partial load, representing high thrust coefficient conditions. We classified distinctive wake paths with reference to yaw misalignment, based on the nacelle wind vane, in steps of 3° in a range of ±10.5°. All paths observed in the nacelle frame of reference showed a consistent convergence towards 0.9 rotor diameters downstream suggesting a kind of wake deviation shift. This contrasts with published results from wind tunnels which in general report a convergence towards the rotor. The discrepancy is evidenced in particular in a comparison which we
performed against published paths obtained by means of tip vortex tracking.

## 1 Introduction

Lately, attention has been directed towards the effects of wind turbine misalignment on wake aerodynamics. Wind tunnel experiments (e.g. Medici and Dahlberg, 2003) and numerical simulations (e.g. Jiménez et al., 2010; Fleming et al., 2014) suggest that the steady path followed by a wind turbine wake can be regulated actively. Consequently, it is envisioned to
optimise the wake path of individual wind turbines to increase the overall efficiency of a wind farm. One of the techniques proposed to achieve this relies on the active control of the wake lateral position by means of active inducement of rotor yaw misalignment. The aim is to avoid full wake effects for wind directions of perfect turbine alignment by means of control strategies taking into account the global wind farm performance, see for instance the work of Gebraad et al. (2016) and Fleming et al. (2016). At present, the question arises regarding the transferability of the findings of the controlled lab and
numerical experiments to the full field. The experimental study of the deviation effect is not simple since the wake flow has to be measured comprehensively in time and space. Therefore, special measuring techniques have to be applied in order to resolve the flow field in a spatial domain covering substantial part of the wake to finally estimate the wake position along its downstream path.

In wind tunnel experiments the quantification of the wake path is performed by means of tip vortex tracking (e.g. Grant et al., 1997; Medici and Dahlberg, 2003; Haans et al., 2005). The technique is based on visualisation of passive tracers revealing the deviation of vortex generated at the blade tips. Additionally, techniques which reveal the full wake flow using particle image velocimetry (PIV) are also applied (e.g. Krogstad and Adaramola, 2012; Rockel et al., 2014). In spite of their suitability in the lab, these visualisation techniques are, to our knowledge, not applicable at the scales of modern commercial wind turbines. A full field alternative technique is the application of remote sensing techniques. Together with other researchers the first author of this paper proposed a wake tracking technique in Trujillo et al. (2011) which is based on two-dimensional lidar measurements from the wind turbine nacelle. In that research the technique is applied in the far wake at four diameters (4 $\mathcal{D}$) downstream of a medium scale wind turbine. Others have applied a similar methodology, of fitting a predefined wake shape to lidar data, in order to capture the far wake path of large-scale wind turbines (e.g., Aitken et al., 2014; Machefaux et al., 2015). However, we have not found literature about experimental studies in the near wake between the rotor and 2 $\mathcal{D}$ downstream relating to yaw misalignment effects. Such measurements are useful to understand the wake behaviour in its full extent and for validation of findings with physical and numerical models. Moreover, such measurements should give insight into the relation of measurements in the vicinity of the rotor, for instance the nacelle wind vane, and the wake deviation. This is helpful regarding the future needs in sensors for implementation of active wake control strategies.

In this research we aim to measure horizontal near wake paths and observe the dependence of their trajectory on yaw misalignment, by applying nacelle based wake tracking techniques. The investigation is based on data processed for a proof-of-concept presented orally at the conference of the European Wind Energy Association in March 2014 under the title "Measuring wind turbine yaw misalignment by wake tracking" and with the same authorship of this paper. Initially we revise a wake tracking technique already used in far wake studies. Then, we show statistics of wake position tracked between 0.6 and 1.4 $\mathcal{D}$ downstream of an offshore wind turbine. In a next step, we estimate wake paths and classify them with respect to yaw misalignment conditions arising in normal operation. Furthermore, we discuss the factors that may affect the measurement technique and the results. Finally, we compare the findings against results from a published wind tunnel experiment applying vortex tracking in a region from 0.2 $\mathcal{D}$ until 0.5 $\mathcal{D}$ downstream.

## 2  Methods

### 2.1  Near wake path reconstruction from wake tracking

The path followed by the wake of a wind turbine can be reconstructed by estimating the transversal position of the wake deficit at a discrete number of downstream stations. At present, this can be performed in the full field by applying wake tracking on wind fields obtained with nacelle based scanning lidars. In this method the bulk transversal location of the wind speed deficit is identified at each downstream station. For this, the wind velocity is measured on sections parallel to the turbine rotor at defined downstream stations. Figure 1 shows this with the convention for positive yaw misalignment. There, the subscript $l$ stands for the lidar frame of reference, which is attached to the measurement system. This frame of reference can be assumed equal to a nacelle frame of reference if a perfect alignment between the lidar and the nacelle is assumed. Afterwards, the wake offset

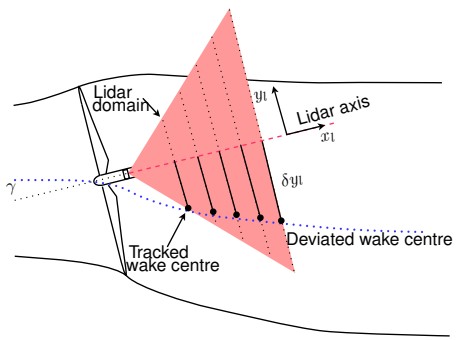 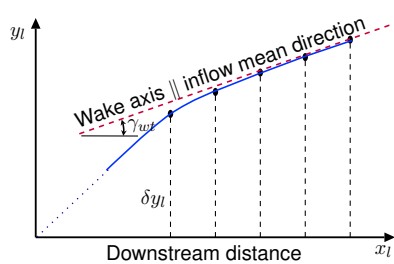

**Figure 1.** Sketch of near wake path reconstruction. (left) Top view of lidar wake tracking. Shadowed area shows the area which is covered by a lidar system aligned with the rotor axis. Perpendicular dotted lines represent one-dimensional or two-dimensional scans parallel to the rotor. Dots depict estimated wake centres. (Right) illustrates Reconstruction of near wake path and estimation of yaw misalignment based on discrete measurements. Lidar frame of reference (subscript $l$) assumed perfectly aligned with the nacelle axis.

in lateral direction ($\delta y_l$) is estimated at each station. The average over a defined period of time of the wake offsets, at each downstream position, describes the near wake path.

## 2.2 Lidar wake tracking with bi-variate Gaussian templates

Several procedures can be applied to estimate a wake centre position based on lidar measurements. For instance, with image
processing techniques the wake limits, and consequently a wake centre, can be identified by using a wind speed threshold defining the inner and outer part of the wake, such technique has been applied by España et al. (2011) on PIV measurements in the wind tunnel. Another approach is to fit the measured wind field to an expected shape of the wind speed distribution across the wake. The selected shape includes a wake position marker among other parameters with or without physical meaning. We find two examples of this, namely the approach by Trujillo et al. (2011), which objective is to find the best fit between a
template function and the wind speed distribution; and the approach applied by Aitken et al. (2014), where the best fit of a flow model is searched.

     In this research we employed a slightly modified version of the wake tracking technique from Trujillo et al. (2011). That method has been developed for analysis of two-dimensional lidar measurements in the far wake. The main difference with the published research is that we used here the non-simplified fitting function and formulated an additional function for more
complex flow. In the following we re-introduce the method and illustrate the additions we make. In the tracking procedure, wind field snapshots are taken across the wake on vertical planes parallel to the rotor. Next, wind speed deficits are calculated by subtracting the average inflow vertical profile measured at an additional system, such as a meteorological mast or forward looking lidar. This procedure intends to isolate the deficit in order to reduce the asymmetry created by the vertical shear and make it more suitable for the final process, where a bi-variate Gaussian function (equation 1) is fitted by means of least-squares
to each wind speed deficit.

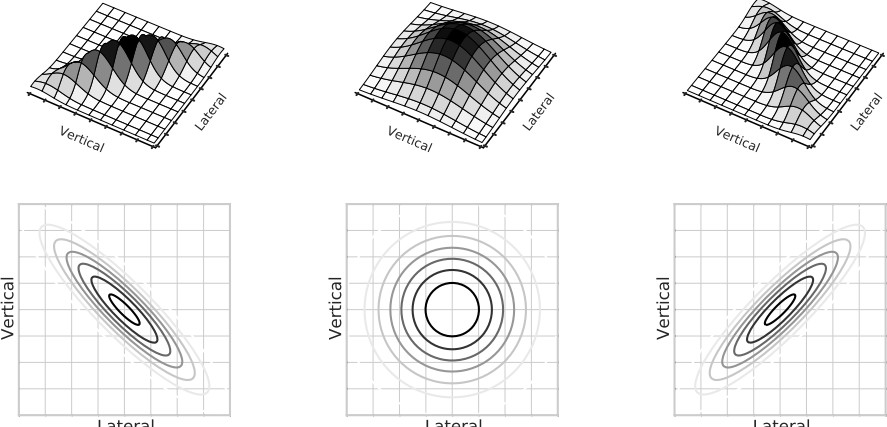

**Figure 2.** Example of bi-variate function (equation 1) for equal parameters $\mu_y$, $\mu_z$, $\sigma_y$, $\sigma_z$ and $A$, but varying correlation values as $\rho = -0.9$, $\rho = 0$ and $\rho = 0.9$, respectively from left to right. Top represents the wake speed deficit and bottom the contour lines.

$$f(y_i, z_i | \mu_y, \mu_z, \sigma_y, \sigma_z, A, \rho) = \frac{A}{2\pi\sigma_y\sigma_z\sqrt{1-\rho^2}}$$
$$\cdot \exp\left[-\frac{1}{2(1-\rho^2)}\left(\frac{(y_i-\mu_y)^2}{\sigma_y^2} - \frac{2\rho(y_i-\mu_y)(z_i-\mu_z)}{\sigma_y\sigma_z} + \frac{(z_i-\mu_z)^2}{\sigma_z^2}\right)\right] \quad (1)$$

The Gaussian function is selected as a fitting template due to its flexibility to adapt to axial-symmetric or non-axial-symmetric shapes. This is convenient in the near wake since the quasi-instantaneous velocity across the rotor is characterised by a complex shape. Furthermore, in the case of analysis of steady wake profiles this function should describe better the expected wake speed profile for different downstream distances; see for instance a review of the wake fields obtained by means of numerical and wind tunnel experiments documented by Vermeer et al. (2003). The target is to obtain the best estimate of $\mu_y$ and $\mu_z$ which define the axis of the Gaussian function. This is assumed to be a marker of the wake centre location in horizontal ($y$) and vertical ($z$) direction. The parameters $\sigma_y$ and $\sigma_z$ characterise the shape lateral and vertical width, respectively. Moreover, $\rho$ represents the correlation between both coordinates and its magnitude ($|\rho|$) is lesser than 1. In the case that the axes of elongation and compression of the shape are aligned with $y$ and $z$ then $\rho$ is 0, otherwise this parameter generates a rotation of the axes (figure 2). We expect this parameter to be useful in capturing non axisymmetric wake shapes, for instance those attributed to wake rotational effects. Furthermore, it is expected that this shape fits in a robust manner with deficits with complex shapes due to uneven induction, rotational effects and boundary layer interaction.

Additionally, the near wake flow is characterised by the detailed rotor aerodynamics. Specially the low induction at the blade root lets wind pass the rotor almost unaffected. This is expressed in a wind speed peak in the near wake centre which we propose to capture by subtracting a concentric Gaussian function from equation 1. The resulting function can be seen in

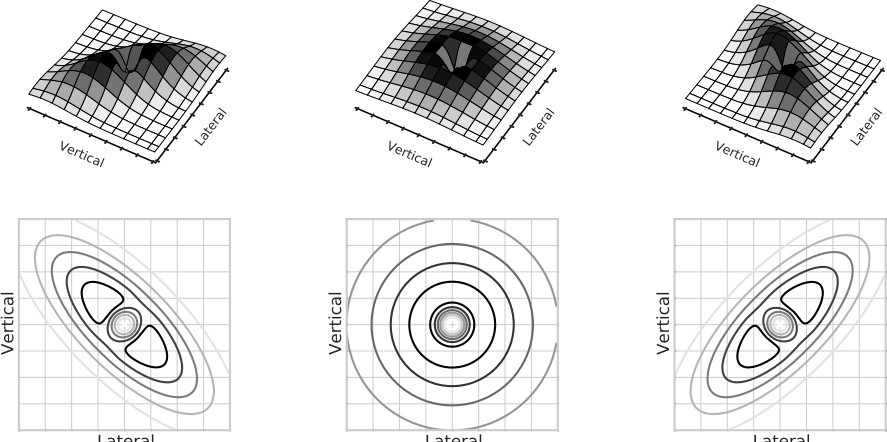

**Figure 3.** Example of extended bi-variate function (equation 2) for equal parameters $\mu_y$, $\mu_z$, $\sigma_{y1}$, $\sigma_{y2}$, $\sigma_{z1}$, $\sigma_{z2}$, $A_1$ and $A_2$, but varying correlation values as $\rho = -0.9$, $\rho = 0$ and $\rho = 0.9$, respectively from left to right. Top represents the wake speed deficit and bottom the contour lines.

equation 2, where the second term on the right hand side (subscript 2) is subtracted from equation 1 (subscript 1). This creates a depression at the highest values of the single Gaussian form describing the wake deficit (See figure 3).

$$f(y_i, z_i | \mu_y, \mu_z, \sigma_{y_1}, \sigma_{y_2}, \sigma_{z_1}, \sigma_{z_2}, A_1, A_2, \rho_1) = \frac{A_1}{2\pi\sigma_{y_1}\sigma_{z_1}\sqrt{1-\rho_1^2}}$$

$$\cdot \exp\left[-\frac{1}{2(1-\rho^2)}\left(\frac{(y_i-\mu_y)^2}{\sigma_{y_1}^2} - \frac{2\rho(y_i-\mu_y)(z_i-\mu_z)}{\sigma_{y_1}\sigma_{z_1}} + \frac{(z_i-\mu_z)^2}{\sigma_{z_1}^2}\right)\right]$$

$$-\frac{A_2}{2\pi\sigma_{y_2}\sigma_{z_2}}\exp\left[-\frac{1}{2}\left(\frac{(y_i-\mu_y)^2}{\sigma_{y_2}^2} + \frac{(z_i-\mu_z)^2}{\sigma_{z_2}^2}\right)\right] \quad (2)$$

The two functions presented here show a formal similarity with the approach applied by Aitken et al. (2014), although extended to two dimensional measurements. However, both procedures differ in principle since our method does not strive to reproduce the model behind the measurements but to identify the existence and location of a bell-like shape. This allows us to even perform tracking directly on line-of-sight measurements without the need of reconstructing the full wind field. This has been tested in a computational setup with large eddy simulation, actuator line for turbine simulation and lidar simulator in Trabucchi et al. (2012). Namely, the wake of a model of a 2 MW wind turbine has been scanned at 2.5 $\mathcal{D}$ downstream with nacelle-based and ground-based lidars. The results showed for the simulated test case, that the wake tracking applied on line-of-sight wind speeds predicts similar wake position than the reference tracking based on the full wind field with average deviations

below 1.5%. Furthermore the study showed a robust behaviour for large misalignments between the mean line-of-sight and the mean wind direction ranging from 0° to 60°.

## 3  Full field experiment at «alpha ventus» offshore wind farm

We applied the techniques for near wake path reconstruction to measurements performed on a wind turbine at the offshore test wind farm «alpha ventus». The measurement campaign took place from the 3rd of March 2011 until the 25th of July 2011. Measurement data have been selected and synchronised as ten-minute averages from three different measurement systems. Namely, the turbine data were obtained from the supervisory control and data acquisition (SCADA) system, the lidar scanner provided the wake wind field data; finally, the inflow data are recorded from a meteorological mast located at the offshore research platform FINO1.

### 3.1  Wind turbine and nacelle based lidar system

The studied turbine is the AV07 which is located in the third row from top to bottom and on the westerly side (shown as a square in figure 4). This turbine is of type Adwen AD 5-116, formerly called M5000-116. It has a rotor diameter ($\mathcal{D}$) of 116 m and a hub-height of 90 m. The cut-in wind speed is 3.5 m/s, the nominal wind speed is 12.5 m/s and the cut-out wind speed 25.0 m/s. Wind turbine data were provided by the SCADA system with a one-minute sampling rate. Mainly, we recorded the nacelle wind vane as the yaw misalignment sensor and estimate a proxy status signal. We estimated this status as an on/off signal by evaluating ten-minute averages of power production and generator rotational speed. Finally, we processed the data in blocks of ten minutes without checking yaw manoeuvres of the turbine in each period of time. We took the decision to skip this check because of practical reasons; for instance, this would force us to work with data blocks shorter than ten minutes and variable in size leading to issues of inflow steadiness with respect to the reference meteorological mast.

The wind turbine near wake has been measured with a research scanning lidar. The system is equipped with a commercial short range pulsed lidar of type Windcube WLS-7 version 1.0 from the manufacturer Leosphere.This is coupled to a scanner with two degrees of freedom, which has been designed by the research group Stuttgart Wind Energy at the University of Stuttgart. The lidar was located on top of the nacelle on the rear part of the helicopter deck of the AV07, as can be observed in figure 4. The position of the source of the lidar, that is the pivot of the scanner mirror, has been estimated as $x =$4.3 m, $y =$-1.1 m and $z =$3.5 m on the nacelle frame of reference. This is a right-hand frame which is centred at the intersection of the tower and the rotor axis and whose $x$-axis points downstream. We estimated an orientation error in the alignment of the lidar scanning axis with reference to the nacelle longitudinal axis is in the order of $\pm 1°$. More details of this system can be found in Peña et al. (2013, chap. 8).

The lidar system measures at five stations downstream, namely, $0.6\mathcal{D}$, $0.8\mathcal{D}$, $1.0\mathcal{D}$, $1.2\mathcal{D}$ and $1.4\mathcal{D}$, which here are numbered from 1 to 5, respectively. The scanning is performed in a quasi-simultaneously process; the scanner described continuously a pattern similar to a Lissajous figure projected into a regularly spaced grid of 7 by 7 nodes. The line-of-sight wind speed was measured simultaneously at the five stations for each scanner position. In effect, the speed of aerosols along the laser beam

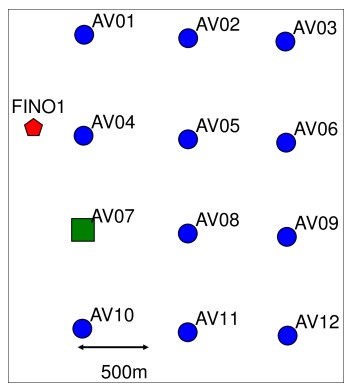
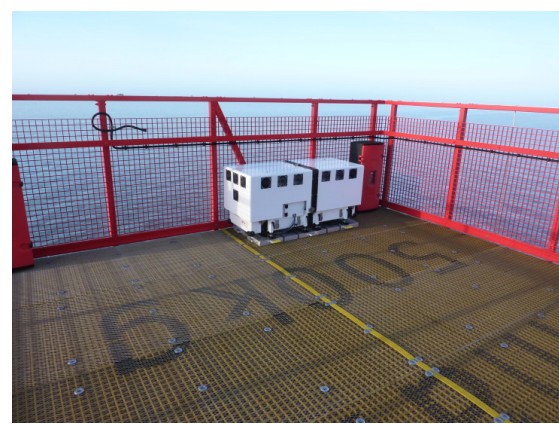

**Figure 4.** Measurement system installation at alpha ventus. Layout of alpha ventus wind farm and position of FINO1 meteorological mast (left). Photography of installation of Windcube plus scanner at helicopter deck of the Adwen's AD 5-116 (right).

is estimated by evaluation of the Doppler frequency shift of back-scattered light at the interrogated positions. The laser operated with a pulse length of $200{\times}10^{-9}$ s and the processing was performed with a range gate width of $100{\times}10^{-9}$ s (see appendix A). With this setup the probe length is approximately 34 m. The scanner movement and the lidar were setup to perform a fly-by measurement at each of the grid points. Scanning of flat planes parallel to the wind turbine rotor is achieved by updating the

5 lidar ranges at each grid point continuously. To cover the measurement grid the angles in elevation and azimuth of the scanner were always below 20°; this means that for wind turbine yaw misalignment in normal operation, ranging from -10° until +10°, we expect a maximum of approximately 30° misalignment between the wind vector and the line-of-sight. Figure 5 shows an example of a measurement performed over approx. 9 s. The intensity map represents the line-of-sight wind speed interpolated by means of a Delaunay triangulation on a grid with higher spatial resolution than the measurement grid. We performed wake

10 tracking by fitting the template function (equations 1 and 2) on such measured fields. This was done applying a trust region algorithm [1] which has the ability to perform bounded optimisation and in this way the domain of valid wake centre positions has been restricted to not exceed one rotor radius ($|\mu_y| \leq \mathcal{R}$ and $|\mu_z| \leq \mathcal{R}$).

### 3.2 Inflow conditions from meteorological mast at FINO1 platform

The windfarm inflow wind conditions for westerly winds were provided by the offshore meteorological mast FINO1. This

15 is located to the north-west of the AV07 and is separated by a distance of approximately 8$\mathcal{D}$ (figure 4). Wind speed vertical profiles are measured with calibrated cup anemometers at heights above the lowest astronomical tide (LAT) of 34.0 m, 41.5 m, 51.5 m, 61.5 m, 71.5 m, 81.5 m, 91.5 m and 103.0 m. Additionally, three-dimensional sonic anemometers provide also wind speeds at three heights, namely, 41.5 m, 61.5 m and 81.5 m. These two types of anemometers are located at different sides of the meteorological mast presenting different behaviour with respect to wake effects from the mast. To obtain suitable

---

[1]Specifically we have used a trust region reflective algorithm implemented in the Optimisation Toolbox (Version 6.4) of Matlab©: Version: 8.2.0.701, The MathWorks Inc., Natick, Massachusetts, 2013.

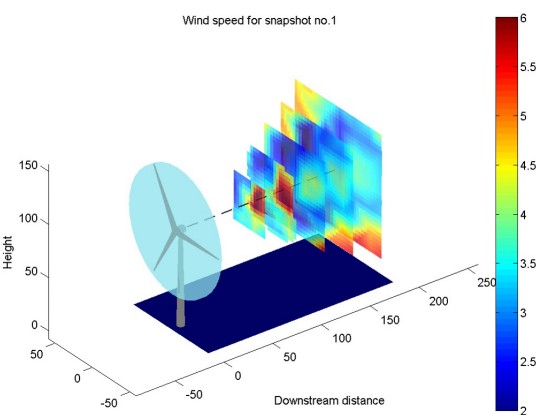

**Figure 5.** Example of snapshot taken with a nacelle based lidar scanner. Measurement taken over approx. 9 s. The colour map represents the line-of-sight wind speed in [m/s] interpolated into a high resolution grid.

vertical profiles we processed the corrected cup and non-corrected sonic anemometers independently for each particular wind direction. The uncertainty in these signals is estimated to vary approximately between 2% and 4% based on the information given by Westerhellweg et al. (2010, 2012). Moreover, we obtained the wind direction near hub-height from the wind vane installed at 91.5 m LAT where the uncertainty is estimated to vary between $2°$ to $6°$ (Westerhellweg et al., 2012). We selected data sets for wind directions where the turbine operates undisturbed by any of the neighbouring wind turbines, namely for wind directions in the sector between $209°$ and $330°$. These angles are given in meteorological convention with $0°$ pointing to north and positive angles in clock-wise direction. For these, conditions the average turbulence intensity, calculated on the basis of ten-minute data as $I_o = \sigma_u/\bar{u}$, was $I_o = 0.165$ with a standard deviation of 0.049.

Finally, we performed an extrapolation of the vertical profile to estimate wind speeds above hub-height and until the highest height measured by the lidar. This was performed using the measurements from 34.0 m until 91.5 m LAT. For this, we assume that the profile follows the Businger-Dyer model for the diabatic boundary layer profile $u(z|u_*, z_o, L)$ as presented by Stull (1988). The model parameters, friction velocity $u_*$, roughness length $z_o$ and Monin-Obukhov length $L$, were estimated for each ten-minute time series by performing a least-squares fit to the ten-minute averages measured at FINO1.

A classification of the data with respect to atmospheric stability has been omitted since the obtained parameters presented high uncertainty. In effect, the results for $L$ showed large variations and, in some cases, contradictory stability classification of consecutive data sets. Two factors can be the main origin of this, first, we used a short averaging period of ten minutes; second, there was a limitation in the measurement setup since the lowest measurement height was too high (above 30 m). These factors have been exposed by Cañadillas et al. (2011) who have analysed in detail stability effects at FINO1. Their analysis was performed with sonic anemometer data and the more elaborated co-variance method. Nevertheless, they found a high scatter of the results when comparing against a bulk Richardson approach including mean meteorological measurements of 30 minutes and sea surface temperature data.

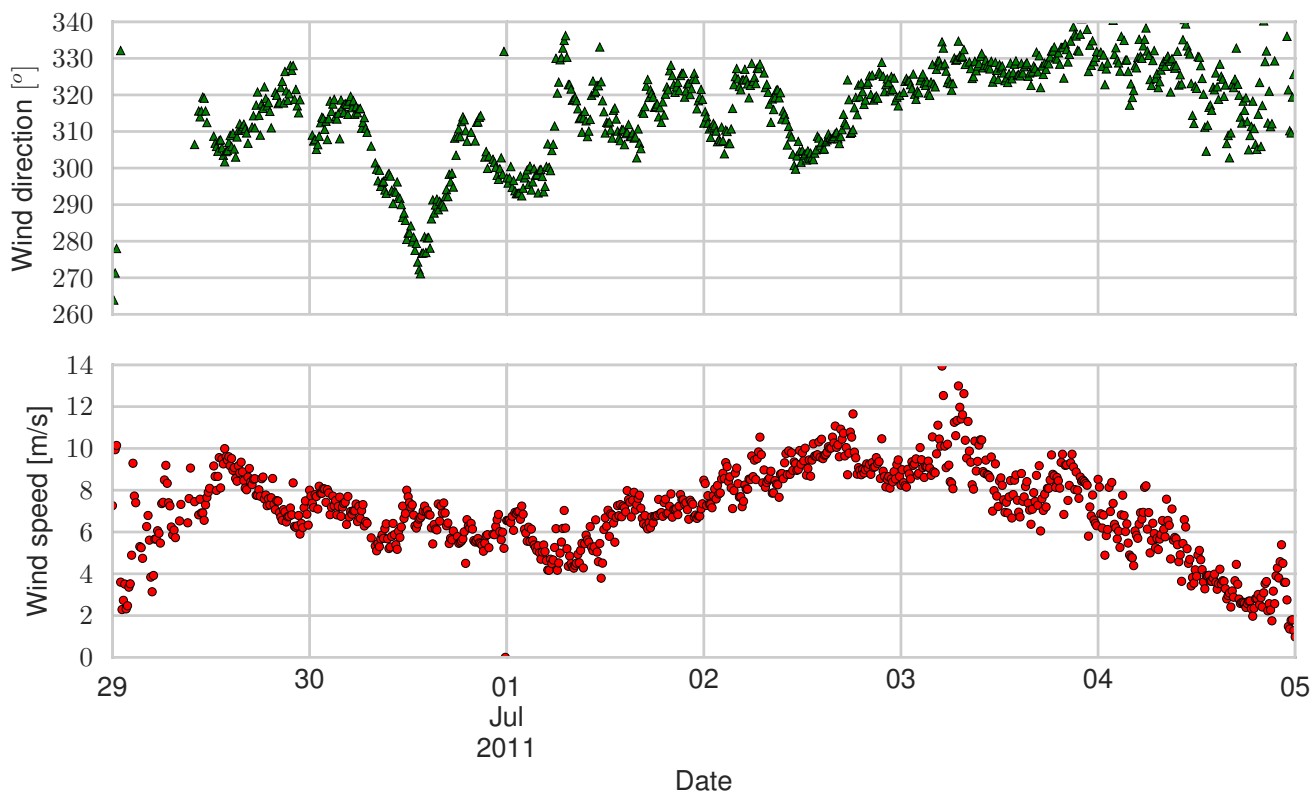

**Figure 6.** Inflow conditions: measured wind speed and wind direction at FINO1 at a height of 91.5 m LAT during selected period from the 29th of June 2011 at 00h UTC until the 5th of July 2011 at 00h UTC.

## 4 Results

### 4.1 Wake position statistics from selected single Gaussian tracking

During this experiment, we selected measurements for six consecutive days from the 29th of June 2011 at 00h UTC until the 5th of July 2011 at 00h UTC. During this period of westerly winds the wind turbine operated in free flow and mostly
5   under wind speeds below rated speed, as can be seen in figure 6. This means that the data are representative for partial load conditions, where the thrust coefficient curve is relatively constant with values estimated to be above $c_T = 0.77$ during 75% of the analysed time. Additionally, in regards to the yaw control of the turbine, during this time the turbine performs maximum two yaw corrections of some degrees in a ten-minute period.

We selected the lidar data for the same dates and wind direction, furthermore we analysed them in ten-minute blocks
10   containing approximately 65 wind field snapshots. These data sets were checked concerning the back-scattered signal quality by filtering out low quality data not complying with the lowest acceptable value of carrier-to-noise-ratio CNR =-17 dB.

Next, we tracked the wake position in each of the measured fields. First the wind speed was reconstructed by assuming full alignment of the wind vector with the rotor axis. Afterwards, the wake wind speed deficits were calculated by subtracting the inflow wind speed profile. This was obtained by averaging over ten minutes measurements at the FINO1 meteorological mast and performing extrapolation as explained in section 3.2.

Some tests have been performed with the extended template function in equation 2; mainly this function has been applied to a selected ten-minute time series using the simplification that $\sigma_{y_2} = \sigma_{z_2}$. A qualitative comparison of the measured line-of-sight wind fields against the wind fields calculated with the fitted template functions showed better agreement of the double Gaussian than the single Gaussian template function. However, the obtained wake offsets were similar for both functions. Although the wake tracking with the double Gaussian template function was expected to be more robust in the nearer stations 1 and 2, we did not find a noticeable improvement in the convergence of the fitting algorithm for those stations. Due to this fact and an expected higher computational cost of the extended function, we applied the simpler function.

After performing wake tracking with this template, we selected ten-minute time series of $(\mu_y, \mu_z)$ where at least 70% of the snapshots have been tracked successfully at each downstream station. This means that for that percentage of scans the fitting process has converged with the given bounds for all parameters and the convergence criteria. We have defined this ad-hoc limit value to deal with unsuccessful fits. Primarily, we have identified three main sources of failed fitting attempts namely, a highly complex wind field, numerical issues of convergence and nonexistence of wake deficit. The last one is related to the operational status of the turbine which at the time of applying the tracking procedure was not available. The results of sampled tests suggested a rather low effect of the first two sources, therefore we assume that any large lost of tracked centres should be revealing a downtime of the turbine. In conclusion the 70% is expected to guarantee that in a ten-minute period the turbine was under normal operation during that percentage of time. Afterwards, we calculated the ten-minute averages $(\overline{\mu}_y, \overline{\mu}_z)$ at each station. Finally, for reconstruction of average wake paths, we selected only ten-minute blocks for which data is available at all five stations and which complied with the inflow and signal quality criteria mentioned before. For this experiment, we found 348 ten-minute data sets complying with all these criteria. This is equivalent to $\sim$51% rate of success taking into account that 678 ten-minute data sets complying with the wind direction sector requirement.

The statistics of wake offset $(\delta y_l)$ for the selected data are shown in table 1. A linear regression through the mean values has given the function $\delta y_l = 5x - 5$, where $\delta y_l$ is given in meters and $x$ in $\mathcal{D}$. The slope is positive and equivalent to $2.6\,^\circ$. The data distribution can be seen in figure 7, where violin plots show the frequency distribution of wake offsets at each downstream station plotted together with the first, second and third quartiles.

Additionally, we have made an estimation of the wake skewness, for the operational conditions of this experiment, whose typical values should lie below $3^\circ$ as shown in appendix B.

## 4.2   Near wake path dependency on yaw misalignment

The dependency of the mean near wake path on yaw misalignment is shown in figure 8. This shows the results of classifying paths with respect to yaw misalignment based on the nacelle wind vane measurements. We have set an ad hoc bin size of $3\,^\circ$

**Table 1.** Statistics of wake offset obtained with simple Gaussian tracking for selected week. All values in lidar frame of reference given in [m] but the count.

|        | $\delta_{y_1}$ | $\delta_{y_2}$ | $\delta_{y_3}$ | $\delta_{y_4}$ | $\delta_{y_5}$ |
|--------|------|------|------|------|-------|
| count  | 348  | 348  | 348  | 348  | 348   |
| mean   | -1.8 | -0.4 | 0.2  | 1.2  | 2.7   |
| std    | 4.6  | 1.8  | 1.9  | 3.4  | 5.3   |
| min    | -12.1| -4.5 | -4.8 | -8.2 | -10.0 |
| median | -1.7 | -0.4 | -0.2 | 1.3  | 3.0   |
| max    | 26.5 | 8.4  | 8.4  | 11.2 | 15.8  |

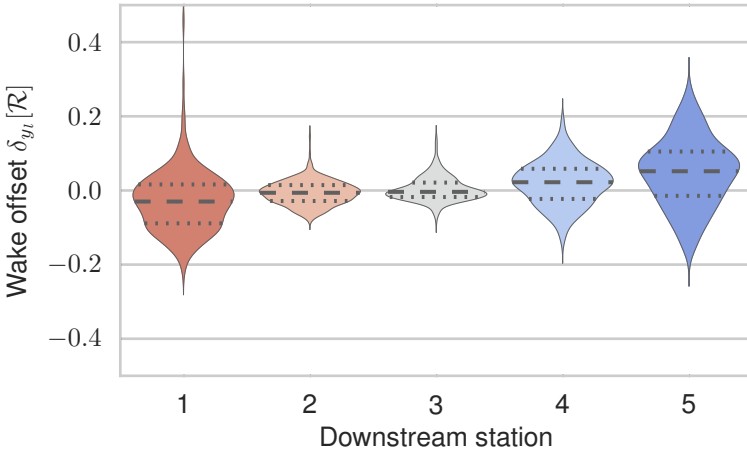

**Figure 7.** Violin plot of statistics showing the distribution of the frequency of the horizontal wake offset. The dashed lines represent the 25th, 50th and 75th percentile. $\delta_{y_l}$ is given in rotor radius $[\mathcal{R}]$.

and performed averaging of all selected data at each downstream distance. The error bars present the standard error on the mean calculated as $s(\overline{\delta_{y_l}}) = s_{\delta_{y_l}}/\sqrt{N}$, where $s_{\delta_{y_l}}$ is the standard deviation and $N$ is the number of data points.

The mean yaw misalignment angles estimated from the mean near wake paths ($\gamma_w$) are shown in table 2. These were calculated by linear regression of the last three downstream stations ($1.0\mathcal{D}$, $1.2\mathcal{D}$ and $1.4\mathcal{D}$), whereby $\gamma_w$ is a transformation
5  of the slope into degrees in the frame of reference of the rotor, namely $\gamma_w = -\gamma_{wt}$ (see figure 1). The difference between the vane and wake tracking based misalignment is calculated as $\Delta\gamma = \gamma_v - \gamma_w$ and its average over all bins is $\overline{\Delta\gamma} = +\,2.7\,°$.

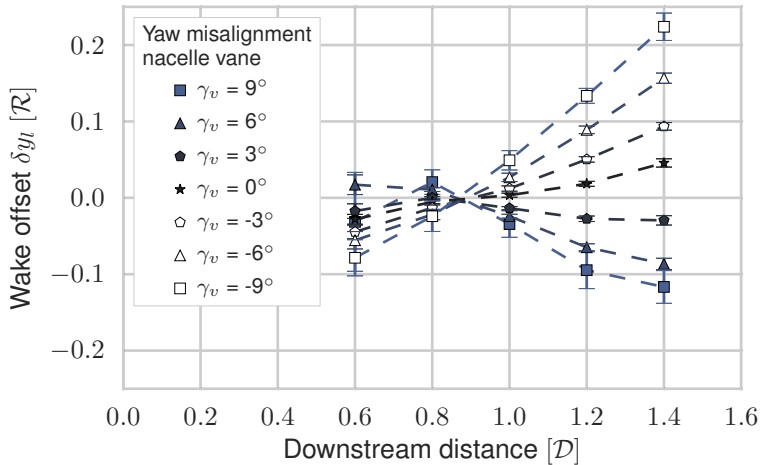

**Figure 8.** Mean near wake paths measured by lidar wake tracking classified in 3° bins with respect to yaw misalignment measured with the nacelle vane. Bars showing standard error on the mean.

**Table 2.** Results of estimation of yaw misalignment from wake tracking ($\gamma_w$) by linear regression through the last three stations of the near wake path (1.0$\mathcal{D}$, 1.2$\mathcal{D}$ and 1.4$\mathcal{D}$). Data classified in 3° bins with respect to yaw misalignment from nacelle vane ($\gamma_v$)

| $\gamma_v$ [°] | $\gamma_w$ [°] | $\Delta\gamma$ [°] | Slope [m/$\mathcal{D}$] | Offset [m] | $r^2$ [$-$] |
|---|---|---|---|---|---|
| +9 | +5.9 | 3.1 | -11.9 | 9.6 | 0.9 |
| +6 | +4.5 | 1.5 | -9.0 | 7.5 | 1.0 |
| +3 | +1.1 | 1.9 | -2.3 | 1.4 | 0.8 |
| 0 | -3.0 | 3.0 | 6.1 | -6.0 | 1.0 |
| -3 | -5.8 | 2.8 | 11.8 | -11.1 | 1.0 |
| -6 | -9.2 | 3.2 | 18.8 | -17.3 | 1.0 |
| -9 | -12.3 | 3.3 | 25.4 | -22.6 | 1.0 |

## 5 Discussion

### 5.1 Characteristics of full field near wake paths

Two main features can be observed in the near wake paths obtained during this experiment (figure 8). Namely, what we call a shift of wake deviation and an asymmetry of the wake paths with reference to the nacelle longitudinal axis. We give the following interpretation to these results.

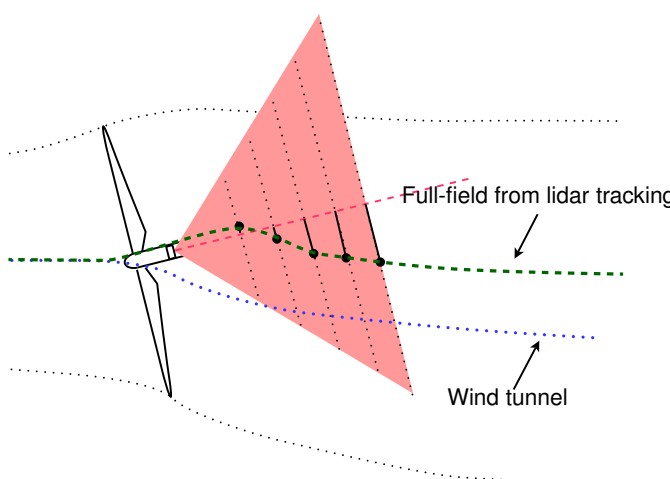

**Figure 9.** Sketch of our interpretation of the measured near wake path for an individual yaw misalignment; showing what we call a deviation shift with respect to wind tunnel paths found in publications.

### 5.1.1 Shift of wake deviation

A feature of the obtained near wake paths is their consistent convergence towards the centre at a downstream distance approximately at $x_l \simeq 0.9\,\mathcal{D}$. We interpret this as if, just after this distance, the wake would deviate from the longitudinal axis of the nacelle as expected from the pure rotor thrust balance; therefore, we call this a wake deviation shift. A representation of this, is presented in figure 9 for one hypothetical path. This contrasts with the wind tunnel observations (e.g., Grant et al., 1997; Haans et al., 2005; Medici and Dahlberg, 2003) which show that the near wake paths meet at the nacelle area or very close to the rotor for horizontal yaw.

This is showing apparent differences in the pattern between the wake flow experienced in the published lab experiments and in our experiment. A hypothesis about this is that the blockage effect of the nacelle could be affecting the deficit in the centre but not at its radial limits. This could in fact lead to different results from both tracking procedures. In conclusion, this evidences a need for studies where the wake path estimation of both tracking techniques can be compared directly.

### 5.1.2 Near wake path asymmetry

In the resulting paths we appreciate two regions, divided approximately at $0.9\,\mathcal{D}$, which show different characteristics of symmetry about the nacelle longitudinal axis. In the region between $0.6\,\mathcal{D}$ and $0.8\,\mathcal{D}$ the paths show a trend to lie towards the right side, when looking downstream. In contrast, at larger distances from $1.0\,\mathcal{D}$ until $1.4\,\mathcal{D}$ the paths are distributed more evenly to both sides. In this region however, there is some apparent and slight bias towards the left side. In spite that we can quantify an average between the yaw misalignment estimated with the wind vane and with the mean wake path ($\overline{\Delta\gamma} = +2.7\,°$), it is not possible for us to differentiate this as a pure, or a combination of, wind vane or lidar wake tracking bias. In the following we contemplate some factors that affect this:

– Horizontally asymmetric mean inflow conditions

A steady horizontal wind shear could be responsible for a constantly deviated wake towards one side. However, we think that the probability of such inflow conditions is very low. Mainly, during the evaluated period of time, the wind direction range selected is relatively wide so that any local effects, due for instance to wind farm geometry, should not be persistent for the whole data set.

– Bias of the nacelle wind vane

We have checked the correlation between two sources of yaw misalignment, namely the nacelle wind vane and the difference between the wind direction measured at the meteorological mast FINO1 and the yaw position of the rotor. After performing a linear regression we obtained a bias of approximately $+3\,°$ for the wind vane, which seems to coincide with the $\overline{\Delta\gamma} = +2.7\,°$.

– Biased wake tracking method

The tracking method is applied here independently to each measured snapshot. In this process, the first guessed position in the minimisation algorithm is $\mu_y = 0$ and $\mu_z = 0$. Therefore, any bias of the method towards one direction could be given by the optimisation algorithm used. However, it does not seem plausible for it to have a predilection for one specific side. First, the function is symmetrical and smooth and second, the optimising algorithm is not expected to have a preferred search direction. Therefore, any bias caused by the lidar wake tracking should be mainly caused by an error in orientation of the system on the nacelle. We have estimated an orientation error in the order of $\pm 1°$ based on our knowledge of the lidar system and the mounting at the nacelle. However, the actual misalignment between the longitudinal axis of the lidar and the nacelle is unknown to us and a deviation from the estimated error could be possible. An accurate verification of this value could be performed through evaluation of nearby hard targets; however, this was practically not possible for us due to the lack of such external objects in the achievable range of our lidar system.

– Interaction of surrounding flow and rotating wake

A plausible hypothesis is that the near wake path could show an asymmetry as a result of the interaction of wake flow rotation, vertical wind shear and wake deficit. Such effect has been observed experimentally in measurements analysed in the far wake in Trujillo et al. (2011).Furthermore, Dörenkämper (2015, p. 117) has made numerical analysis of the effects of atmospheric stability through large eddy simulations, showing that the wake mean deviation is sensitive to the atmospheric stability conditions. Although the mentioned studies do not get into the near wake region, we expect those far wake effects to be also visible in the near region and therefore could be observable in our measurements.

Finally, despite the possible existence of such asymmetry, we expect that the implemented tracking method still will work effectively without generating a bias in the horizontal wake offset. This is due to the low misalignment angles analysed, for which no deformation of the mean wake deficit should be expected. We have not checked this directly but base our assumption on results of wind tunnel experiments. For instance in the work of Krogstad and Adaramola (2012) we recognise that the

shape of the wake changes for large yaw misalignments, yet we do not perceive any deformation between perfect alignment and $\gamma = 10°$.

## 5.2 Comparison against wind tunnel measurements

The shape of the near wake paths measured during this campaign presents features which contrast with measurements of models

in the literature. A qualitative comparison with a published wind tunnel experiment (Grant et al., 1997) has been performed in figure 10. The wake paths from wake tracking have been sorted in bins of $\gamma_v = 5°$ to make them comparable to the published paths. Additionally, the results are also summarised in table 3.

In the wind tunnel experiment, the near wake paths of a turbine model of $0.9\,\text{m}$ diameter have been estimated by means of tip vortex tracking. The tip vortices have been observed and localised by means of the laser-sheet visualisation technique in

an area ranging between $\sim 0.2\mathcal{D}$ and $\sim 0.5\mathcal{D}$. Therefore these results are complementary to the measurements in this paper. In regards to the operational conditions of the model turbine, the thrust coefficient is not documented. However, it is inferred from the reported skewness that the thrust coefficient is relatively large. The wake offsets for the three bladed model have been digitised and further processed for the comparison. First, the wake offsets have been normalised with the rotor radius; and second, they have been mirrored about the $x$-axis due to the application of an opposite convention of yaw misalignment in both

papers.

The most evident difference between both measurements is what we call a wake deviation shift. This phenomenon is not present in the wind tunnel measurements. Additionally, an asymmetry of the wake paths of the model turbine can be identified, however, such behaviour can not be easily stated in the full field paths if the exact bias between yaw misalignment measurement and lidar orientation is not defined, as discussed in the last section. Finally, with the same background it is also not possible to

draw conclusions on differences in wake skewness.

**Table 3.** Results of estimation of yaw misalignment ($\gamma_w$) by linear regression through last three points of the near wake path ($1.0\mathcal{D}$, $1.2\mathcal{D}$ and $1.4\mathcal{D}$). The data have been classified in $5°$ bins with respect to nacelle vane ($\gamma_v$)

| $\gamma_v$ [°] | $\gamma_w$ [°] | $\Delta\gamma$ [°] | Slope [m/$\mathcal{D}$] | Offset m | $r^2$ - |
|---|---|---|---|---|---|
| 10 | 5.9 | 4.1 | -11.9 | 9.6 | 0.9 |
| 5 | 3.2 | 1.8 | -6.6 | 5.3 | 1.0 |
| 0 | -3.0 | 3.0 | 6.0 | -6.0 | 1.0 |
| -5 | -7.8 | 2.8 | 15.8 | -14.7 | 1.0 |
| -10 | -12.3 | 2.3 | 25.4 | -22.6 | 1.0 |

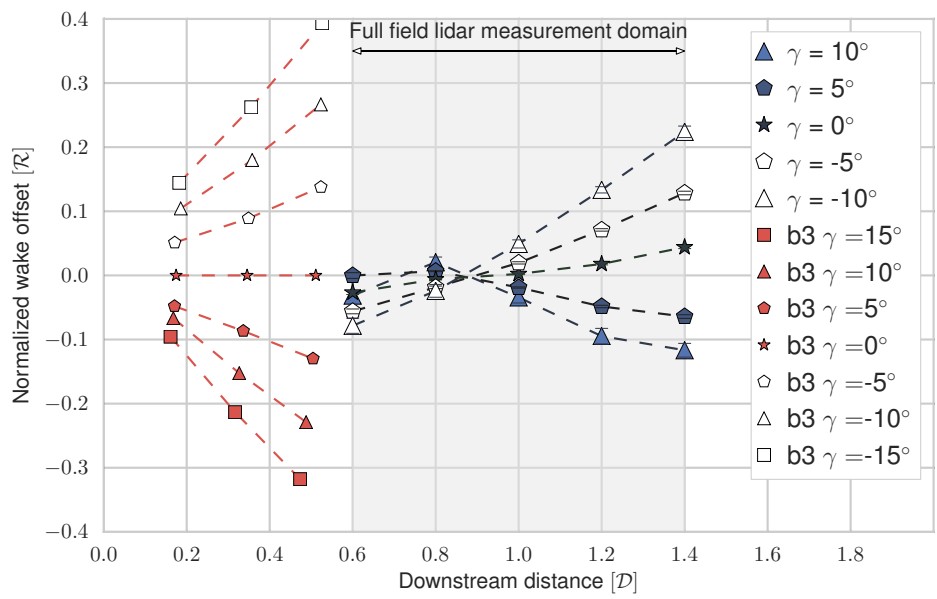

**Figure 10.** Comparison of published wind tunnel and full field mean near wake paths in the frame of reference of the turbine. The wind tunnel paths (from $\sim 0.2\mathcal{D}$ until $\sim 0.5\mathcal{D}$) have been digitised from Grant et al. (1997). Full field paths from lidar averaged in $5°$ bins of yaw misalignment.

### 5.3 Wake tracking performance

The wake tracking method shows a robust behaviour for some measurements. First, the wake position can be estimated successfully even for measurements covering partially the wake cross section. Second, the wake can be tracked also for complex shapes which do not resemble exactly the predefined Gaussian shape. This is of advantage due to the highly complex character of the wake wind field. In spite of this, some limitations are seen in terms of overall availability which amounts to 51.8%. The main reason for this can be evidenced in the results shown in figure 7, where in the first two downstream stations ($0.6\mathcal{D}$ and $0.8\mathcal{D}$) the frequency distribution of wake offset shows larger changes than in the other stations further downstream. This can be attributed to failed convergence of the tracking process, rather than to a physical larger wake deviation at that position. This can be related to numerical issues, as well as to a extremely reduced measured area and a very complex flow field. As can be observed in figure 5 the unaffected flow at the nacelle level is very evident in the first stations and vanishes towards the end of the measurement domain at $1.4\mathcal{D}$. In this case, the nearer downstream stations 1 and 2 are affected in a major degree than stations 3, 4 and 5, which show almost no effect by this complex flow. This suggests that to solve this problem the scanning of the very near wake must cover a larger area.

## 6 Conclusions

In this research we aimed at measuring in full scale the horizontal path followed by the wake of a wind turbine; and observe the effects of yaw misalignment on their trajectory in a region near the rotor which has not been documented in the literature. We applied successfully a lidar wake tracking technique that has been tested previously to obtain far wake paths. We have analysed the near wake of a 5 MW offshore wind turbine for distances between $0.6\mathcal{D}$ and $1.4\mathcal{D}$ and under yaw misalignment arising in normal operation under free inflow conditions.

We identify a general limitation in the application of nacelle based lidar for the analysis of the very near wake. Mainly, the performance of wake tracking is strongly diminished at the nearest distances to the rotor due to the reduced area which is interrogated by a lidar scanner. Despite the restrictions, we were able to obtain plausible full paths with a success rate slightly above 50%. Furthermore, we succeeded in classifying distinctive wake paths with respect to yaw misalignment in angular bins of $3\,^\circ$ in the range of $\pm 10.5\,^\circ$. These results suggest that, letting aside the bias, a consistent correspondence can be found between the tracked wake positions and yaw misalignment observed by the nacelle wind vane.

We appreciate an apparent asymmetry in the wake paths in reference to the nacelle longitudinal axis. Nonetheless, it is not possible for us to conclude whether this is due to orientation errors of the instruments or to actual systematic asymmetry of the wake paths. A more accurate estimation of lidar positioning and yaw misalignment than in our campaign is needed to disambiguate this issue.

In regards to the comparability against published results obtained in wind tunnel studies, we show that the wake paths are qualitatively different. A clear difference is that in our measurements all wake paths, in the lidar frame of reference, show a consistent convergence towards a downstream distance of approximately $0.9\mathcal{D}$. We interpret this as a wake deviation shift that has not been reported in wind tunnel studies.

Finally, the differences that we identify between the lab and our measurements lead us to conclude that there is need of research comparing wake paths obtained through tip vortex and wake tracking techniques. We think that such work can give insight into the observed behaviour of our wake paths and would explain the discrepancies that we have found in this research.

*Acknowledgements.* We thank the support of Adwen for the support during the setup of the measurement campaign. We are also grateful to Wilm Friedrichs and Frederik Berger at the University of Oldenburg for their proofreading and their valuable suggestions for improvement of this paper.

This research makes part of the German joint project «OWEA Loads - Probabilistic load description, monitoring and load reduction of future offshore wind turbines». It is supported by The Federal Ministry for Economic Affairs and Energy of Germany (BMWi) under contract No. 0325577B.

## Appendix A: Characteristics of lidar-scanner system

Table 4 show the specifications of the lidar scanner system used for the measurements in this research.

**Table 4.** Characteristics of lidar scanner of University of Stuttgart. Windcube WLS-7 version 1, extended with a scanning unit.

| Property | Value |
| --- | --- |
| Wavelength | $1.54 \times 10^{-6}$ m |
| Repetition rate | 10000 Hz |
| Pulse Energy | $10 \times 10^{-6}$ J |
| Pulse length (FWHM) | $200 \times 10^{-9}$ s $\rightarrow$ 30 m |
| Measurement range | 40 m to 200 m |
| Number of range gates | 5 |
| *Data acquisition* | |
| Photo diode sampling rate | $250 \times 10^{6}$ Hz |
| Range gate width | $\sim 0.1 \times 10^{-6}$ s $\rightarrow$ 15 m |
| *Scanning* | |
| Scanner | two degrees-of-freedom |
| Target points trajectory | 49 |
| Period full trajectory | $\sim 9.125$ s |

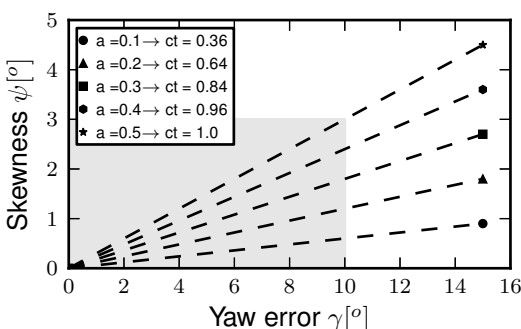

**Figure 11.** Representation of the effect of wind turbine misalignment ($\gamma$) on wake skewness ($\psi$).

## Appendix B: Wake skewness

From theoretical considerations the wake deviation ($\chi_o$) can be approximated as $\chi_o \simeq (0.6a + 1) \cdot \gamma$ (see vortex model in Burton et al. (2001)) what derives into a skewness equal to $\psi \simeq 0.6a\gamma$. This behaviour is depicted in figure 11 , where an approximate value of thrust coefficient $C_T$ is given for the case of no misalignment. This is done taking into account that based on the vortex model the thrust coefficient is rather unaffected by yaw errors in the order of magnitude shown in figure 11.

The shaded area depicts approximately the yaw misalignment values which are covered during this experiment. As a consequence the highest values of skewness that could be expected are below $\psi \simeq 3°$. However, this value lies already in the order

of the typical uncertainty assumed for nacelle wind vanes. Therefore we conclude that the wake skewness can not be resolved from pure yaw misalignment measurement by the standard wind vane at the nacelle in this experiment.

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
