# Peer review of "Full field assessment of wind turbine near wake deviation in relation to yaw misalignment"

_Wind Energy Science, 2015_

## Referee Comment (RC1) · Anonymous Referee #1 · 23 Jan 2016

The article discusses the near wake deviation of a utility-scale offshore wind turbine as a result of yaw error, which has not been previously studied in the literature. In particular, the authors note a "delay" in the onset of this wake deviation.

Overall, the authors have done some interesting work, and the manuscript is well-organized and concise. Methodologies are clearly explained and figures are used appropriately to enhance the discussion in the text. I approve publication of the article, considering that corrections will be made according to the recommendations given below.

Major comments

Lines 65 and 80: Eqs. (1) and (2) describe the shape of the *wind speed* distribution, even though lidar measurements are later fit to these models and the lidar is only capable of measuring *line-of-sight* velocities. More parameters are needed to account for the azimuth and pitch angles of the lidar beam. This is not a major concern when the lidar angles are small (i.e., less than 10 degrees), but it can introduce a significant source of error for larger angles. I strongly encourage the authors to either (a) update Eqs. (1) and (2) to account for lidar azimuth and pitch, or (b) provide a rigorous justification as to why these angles are small enough to be neglected in this particular experiment.

Line 159: What is the purpose of Eq. (2) if it isn't used? Here, I do not agree with the authors' decision to abandon Eq. (2) in their analysis. Eq. (1) is a function describing the shape of the far wake, while Eq. (2) is a function describing the shape of the near wake. Since the authors make a point of focusing their analysis on the near wake, Eq. (2) is the appropriate fitting function to use in this context. One can easily see from Fig. 5 that Eq. (1) is inadequate for describing the shape of the near wake. I strongly recommend that the authors do the analysis using Eq. (2). It might be harder to employ Eq. (2), but it is the proper equation to use.

Minor comments

Lines 14-17: To support the statements given here, it would be nice if the authors cited any relevant literature surrounding the use of yaw control for wind farm optimization.

Lines 27-29: The use of the first-person pronoun "I" is inappropriate here since the paper under review and Trujillo et al. (2011) both have multiple authors, including the overlap of two authors. I suggest that this sentence begin "Trujillo et al. (2011) developed a wake tracking technique which…"

Line 31-32: Other studies have in fact studied the near wake. For example, Aitken et al. (2014) discusses an experimental study that covered both near wake and far wake measurements. The authors should be more specific by noting that their unique contribution here is their specific focus on near wake deviation due to yaw misalignment.

Line 39 (and elsewhere): I recommend that the authors use the conventional abbreviation "D" for rotor diameter, to avoid spelling out the word "diameter" every time.

Line 60-61: Rather than saying "measured by an additional system" the authors should be more specific and mention that the inflow profile is measured by a met tower upwind of the turbine. Presumably, the ambient profile is not the same upwind and downwind of the rotor, particularly since the met tower is

located almost 1 km upwind of the turbine. This is a deficiency that introduces error into the estimation of the wake properties, which the authors should acknowledge.

Line 66: It is not just that the Gaussian function can be selected for its flexibility. The authors should mention that there are theoretical and experimental justifications for using a Gaussian to fit the wake profile. See, for example, Pope's book from 2000 titled "Turbulent Flows" and:

Magnusson, M., 1999: Near-wake behaviour of wind turbines. J. Wind Eng. Ind. Aerodyn., 80, 147–167, doi:10.1016/S0167-6105(98)00125-1.

Line 69-70: Gaussian functions cannot have a "half-width" since they extend to infinity. It would be more accurate here to mention the full-width at half-maximum (FWHM) or just say that $\sigma_y$ and $\sigma_z$ are parameters that determine the extent of the wake boundary.

Line 70: Magnitude of correlation coefficient can be less than or equal to 1.

Lines 75-80: Eq. (2) is presented without context as though it is entirely original, when in fact it is effectively a two-dimensional analog to Eq. (8) in Aitken et al. (2014). It behooves the authors to acknowledge this previous work.

Line 85 (and elsewhere): Why is the name of the wind farm enclosed in brackets and lowercase? Why is not written as "Alpha Ventus" wind farm?

Line 125: Figure 4 is somewhat misleading since it does not seem to be drawn to scale. The text states that the tower is located 8D or about 900 m from the turbine, but Figure 4 makes it look like the tower is less than 500 m from the turbine. I suggest redrawing the figure to scale.

Line 144: Figure 6 depicts measurements of wind speed and direction. The instruments used to take these observations have some kind of measurement uncertainty. The measurement uncertainty should be quantified in the text, and Figure 6 should be modified to include error bars showing this uncertainty.

Line 156-157: The language here seems vague: "The results looked qualitatively more similar…" What results are being referred to here? More similar in comparison to what? Similarly, the phrase "improvement of success" is also vague. Improvement over what? And what does "success" mean? That the fitting algorithm converged to a sensible solution?

Line 160-161: What does it mean for a snapshot to have been tracked "successfully"? Why is 70% chosen as a cutoff point? This number seems completely arbitrary—are the authors able to justify this cutoff somehow?

Line 167: Is there any significance to 2.6 degrees? Was this close to the average yaw error (the difference between the wind direction measured by the tower and the yaw angle of the turbine) during the experiment? If so, that would be a neat result to point out.

Line 185-187: The wake deviation delay seems analogous to the fact that the maximum velocity deficit is attained 1-2D behind the turbine, as noted in:

J.F. Ainslie. Calculating the field in the wake of wind turbines. Journal of Wind Engineering and Industrial Aerodynamics, 27:213–224, 1988.

and in Sanderse's literature review on the aerodynamics of wind turbine wakes. It would be interesting to point out the similarity between the deviation delay and the velocity deficit since both cases show that, in the real world, the impulse delivered by the rotor on the flow cannot occur instantaneously.

Typos

Line 88: Change "minutes" to "minute".

Line 108: Change "which" to "whose".

Line 195: Change "analog" to "analogous".

---

## Referee Comment (RC2) · Anonymous Referee #2 · 10 Feb 2016

The manuscript entitled "full field assessment of wind turbine near-wake deviation in relation to yaw misalignment" deals with scanning LIDAR measurements in the near vicinity of a wind turbine in order to track the near-wake position and to correlate it with the yaw misalignment conditions of the wind turbine.

The objectives of these measurements are of great interest since the actual influence of a yaw misalignment on the development of near-wake of a full scale wind turbine is still partially unknown since information is generally provided through wind tunnel experiments ( not at the same Reynolds) and numerical computations (high degree of modelling).

The technical challenges of this kind of measurements are very important and the paper describes in a precise manner all the technical parameters necessary to make

the measurement set-up reliable. The method to track the near-wake and the way of filtering data versus the yaw angle are well described.

Consequently, I recommend the publication of the article, considering that corrections will be made according to the recommendations given below.

Major comments:

- The selected site is well documented and the measurement period is long enough to ensure statistically converged results. One could regret that no classification of data versus the thermal stability was performed. Would it be possible to do this classification and observe the consequences on the results? If not, please at least comment on it within the document. What is the average turbulence intensity at hub height?

-$4.1. Knowing the thrust coefficient, it is possible to assess the expected velocity deficit and so, the expected axial induction factor. One can then compare this velocity deficit with the one obtained in the present data base. One can also assess the expected skew angle of the wake, which depends on the axial induction factor and the yaw angle. According to a rapid calculation, the skew angle of the wake should be 1.16 times higher than the yaw angle. However, this skew angle is not taken into account in the present study: why?

- Figure 7: the data scatter at the first position cannot be due to a physical behavior of the near-wake. It is unlikely that the wake changes its position in such a magnitude really close to the rotor. One might conclude that the tracking method based on a Gaussian distribution is not appropriate to capture the very near wake and one could suggest not to interpret results from this very near position

-$4.2. the systematic shift angle of 3° between the wind vane and the wake path sounds as a systematic bias in the vane measurement. This is confirmed in $5.1.2 since a bias of 3° is mentioned. Why not taking this bias into account from the beginning of the data processing in order to remove this systematic shift?
[Figure]

-$5.1.1. "Delay" sounds like a temporal characteristic whereas the authors describe a spatial behavior. The present study does not deal with unsteady properties of the near-wake since the data are time-averaged. "Shift" might be more appropriate in the current situation. This raises also the question of the settling time of the near-wake position after a modification of the yaw angle. In the present study, the history of the misalignment is not taken into account.

Minor comments:

-please indicate the measurement volume (or length?) of the Lidar system : the LIDAR system spatially integrates along a line-of-sight distance, meaning that when you state to measure at a certain position, it is in reality an space average of along a given distance.

- Line 250 : "Vortices "instead of "vortexes"

―――――――――――――――――――――

---

## Author Comment (AC1) · 11 Feb 2016

**Discussion paper wes-2015-5**
**Answers to comments of Referee 1**
**based on document wes-2015-5-RC1-supplement**

Juan José Trujillo et al.

February 10, 2016

**Contents**

**1    Introduction**

Dear Referee,

thank you very much for taking the time to perform a review of this paper and for the constructive comments. Below you will find the answers to your comments. A revised version of the paper including your comments has not yet been created while waiting for comments of additional referees.

Sincerely,

Juan José Trujillo and co-authors.

**2    Major comments**

**2.1    Analysis on line-of-sight data**

Lines 65 and 80: Eqs. (1) and (2) describe the shape of the wind speed distribution, even though lidar measurements are later fit to these models and the lidar is only capable of measuring line-of-sight velocities. More parameters are needed to account for the azimuth and pitch angles of the lidar beam. This is not a major concern when the lidar angles are small (i.e., less than 10 degrees), but it can introduce a significant source of error for larger angles. I strongly encourage the authors to either (a) update Eqs. (1) and (2) to account for lidar azimuth and pitch, or (b) provide a rigorous justification as to why these angles are small enough to be neglected in this particular experiment.

**Answer**

You are right in that we should introduce an explanation on why it is valid to work with line-of-sight data without taking other parameters into account. For this, two points have to be clarified, namely the tracking phylosophy and the effect of misalignment between the line-of-sight and the full wind vector.

- Wake tracking philosophy

  There are several ways to estimate a sort of wake centre position. For instance, one approach could be, as you suggest, based on an expected shape of the wind speed distribution. In this case, the model used to fit the measurements really needs to include several parameters to describe accurately the complexity of the flow situation of the studied case. This is for instance the approach followed by Aitken 2014.

Another approach is the one we use, where the objective is not to find the function describing best the wind speed distribution but, to find any axi-symmetric and smooth function which fits to the wind speed distribution. We use the word 'template' to convey the general sense of the fitting function, which does not have to match perfectly with the wind speed distribution. In effect, we could have used a function different than the Gaussian to fit our data, however, from a practical point of view it is more robust to select a template similar to the expected wind speed distribution if it is known a priori. In consequence, Aitken's and our approaches look similar from a formal point of view although in principle they are completely different. In fact, we can apply equations (1) and (2) directly on the line-of-sight data while Aitken has to elaborate more the model.

It is to note that the outcome of the template approach is mainly the centre of the wake, which was the variable needed for this research. This contrasts with the first approach where other characteristics of the wake such as wake width have a more physical meaning.

- Sensitivity of tracking to misalignment

We have been concerned about the robustness of the tracking method to follow the wake under dynamic conditions. There are two main issues, the first is that in the quasi-instantaneous measurements the resulting wind speed distributions are far from having a Gaussian shape, which is only a characteristic of steady fields, i.e. averaged over a time longer than our sampling rate of one field every nine seconds. The second issue is the one you mention related to the misalignment of the lidar line-of-sight caused by a combination of changing wind direction, lidar scanning and wind turbine yawing.

To approach the question of robustness a numerical study has been performed by means of lidar simulation by Trabucchi et al. 2011. In that experiment the wake of a model of a 2MW wind turbine has been obtained from a large eddy simulation and the turbine has been simulated with the actuator line approach. Due to the computational effort only a test case has been studied. The wake has been scanned with the same scanning pattern as the one used in this research at a distance of 2.5D. Furthermore, the tracking procedure as explained here with a single bi-variate Gaussian function has been applied on line-of-sight data. Both nacelle and ground based lidar simulation results showed robustness for large angles of misalignment well above the 10°

you mention.

Although the turbine in that experiment was smaller, there is no reason to believe that a larger turbine will show different results. Moreover, although only one case was studied, the consistency of the results leads us to assume that the wake tracking with the Gaussian template is not very sensitive to the misalignments experienced during the period evaluated in this paper which were almost all below 10°.

Reference D. Trabucchi, J.J. Trujillo, G. Steinfeld, J. Schneemann, M. Kühn, Simulation of measurements of wake dynamics with nacelle and ground based lidar wind scanners, Wake Conference, Visby, 2011 The proceedings can be downloaded from: `http://space.hgo.se/wake_conference/?q=system/files/bookabstract2011_update_1.pdf`

- Suggestion of correction to the paper

  If you find this sufficient we will make a clearer explanation of the tracking approach and the direct applicability of our approach to line-of-sight measurements in section 2.2.

  Additionally, we will make reference to the misalignment effects and the assumptions we are performing in this respect in section 3.1.

**2.2 Double Gaussian instead of single Gaussian**

Line 159: What is the purpose of Eq. (2) if it isn't used? Here, I do not agree with the authors' decision to abandon Eq. (2) in their analysis. Eq. (1) is a function describing the shape of the far wake, while Eq. (2) is a function describing the shape of the near wake. Since the authors make a point of focusing their analysis on the near wake, Eq. (2) is the appropriate fitting function to use in this context. One can easily see from Fig. 5 that Eq. (1) is inadequate for describing the shape of the near wake. I strongly recommend that the authors do the analysis using Eq. (2). It might be harder to employ Eq. (2), but it is the proper equation to use.

**Answer**

The purpose of presenting Eq. (2) is for completeness to show the self-check that we have performed and that we did not want to let undocumented. In fact we had the same concerns as you in regards to which function should be used as a template. However, due to time constraints we could not perform a full comparison of both methods. Therefore, we took a pragmatic decision

to go for the single bi-variate Gaussian after some tests which showed no significant differences with respect to the estimated wake centre.

The tests indicated us that in the case of the two dimensional snapshots the single Gaussian has a very similar performance as the double Gaussian for our setup. Our interpretation of this result is that the wind speed "bypass" in the wake centre, specially futher than 1D, is not strong enough to make the single Gaussian invalid. This "bypass" effect will depend on the aerodynamic characteristics of the rotor and therefore will change between turbines. Therefore, this result can not be generalised and is particular to the measurements shown in this paper. In fact, our experience with other type of scanning such as by means of PPI (similar to Aitken's one-dimensional PPIs) on other turbines shows that the double Gaussian is really needed for performing an accurate tracking.

**Suggestion of correction to the paper**

We agree with your statement that "Eq. (1) is inadequate for describing the shape of the near wake", however as explained before, our purpose is not to describe the shape but just to track a wake centre. We believe that the investigation shows consistent results from the selected tracking procedure and therefore we still consider valid our decision to use a single Gaussian. We kindly ask you to reconsider your strong recommendation to use Eq. (2) on the whole dataset.

An improvement that we could do is to emphasise that the interchange-ability of single and double Gaussian approach is a particular case for this measurement setup and that it can not be generalised.

**3 Minor comments**

**3.1 Lines 14-17**

To support the statements given here, it would be nice if the authors cited any relevant literature surrounding the use of yaw control for wind farm optimization.

**Answer**

Will be done.

**3.2 Lines 27-29**

The use of the first-person pronoun "I" is inappropriate here since the paper under review and Trujillo et al. (2011) both have multiple authors, including the overlap of two authors. I suggest that this sentence begin "Trujillo et al. (2011) developed a wake tracking technique which. . ."

**Answer**

This is ambigouos and will be corrected

**3.3 Line 31-32**

Other studies have in fact studied the near wake. For example, Aitken et al. (2014) discusses an experimental study that covered both near wake and far wake measurements. The authors should be more specific by noting that their unique contribution here is their specific focus on near wake deviation due to yaw misalignment.

**Answer**

You are right we will be more specific about this.

**3.4 Line 39 (and elsewhere)**

I recommend that the authors use the conventional abbreviation "D" for rotor diameter, to avoid spelling out the word "diameter" every time.

**Answer**

Will be done.

**3.5 Line 60-61**

Rather than saying "measured by an additional system" the authors should be more specific and mention that the inflow profile is measured by a met tower upwind of the turbine. Presumably, the ambient profile is not the same upwind and downwind of the rotor, particularly since the met tower is located almost 1 km upwind of the turbine. This is a deficiency that introduces error into the estimation of the wake properties, which the authors should acknowledge.

**Answer**

At this point in the paper we are explaining the general tracking process and therefore have not made reference to the specific offshore experiment. However, we will make some clarifications in the paper.

The "isolation" of the wake deficit by means of subtracting the vertical profile supports our tracking procedure. It is not necessary for finding the horizontal wake centre position, however it is done to cope with two issues. First, the vertical wind shear breaks the axi-symmetry of the wind field and consequently the convergence of the fitting process, using our selected axi-symmetric template function, can be more difficult. Second, the vertical wake centre position can be biased.

In conclusion, we expect that the inaccuracy in the subtracted vertical wind shear won't have a significant effect on the estimated horizontal wake centre position. Your comment is right in that if we were to extract other wake properties we would have to take more care about the inaccuracies in the measured profile. However, in such case our approach would be unsuitable and we would have to use a wake model including the wind shear as for instance Aitken 2014 did.

**3.6   Line 66**

It is not just that the Gaussian function can be selected for its flexibility. The authors should mention that there are theoretical and experimental justifications for using a Gaussian to fit the wake profile. See, for example, Pope's book from 2000 titled "Turbulent Flows" and: Magnusson, M., 1999: Near-wake behaviour of wind turbines. J. Wind Eng. Ind. Aerodyn., 80, 147–167, `10.1016/S0167-6105(98)00125-1`.

**Answer**

We agree with you however, in the context of our tracking procedure our claim still holds true. We will add the information of the convenience of using a Gaussian as we explained in the major comments.

**3.7   Line 69-70**

Gaussian functions cannot have a "half-width" since they extend to infinity. It would be more accurate here to mention the full-width at half-maximum (FWHM) or just say that $\sigma$ y and $\sigma$ z are parameters that determine the extent of the wake boundary.

**Answer**

You are right, naming it as half-width is not accurate and misleading. We will use a description in the lines of your second suggestion.

**3.8 Line 70**

Magnitude of correlation coefficient can be less than or equal to 1.

**Answer**

We think we get your point, however in Eq. (1) and (2) a value of 1 will lead to a zero division problem.

**3.9 Lines 75-80**

Eq. (2) is presented without context as though it is entirely original, when in fact it is effectively a two-dimensional analog to Eq. (8) in Aitken et al. (2014). It behooves the authors to acknowledge this previous work.

**Answer**

You are right that we should have given more context to this research. In fact the data used for this paper were processed at the end of 2013 and beginning of 2014 in the context of an oral presentation at the European Wind Energy Conference (EWEA) with the title "Measuring wind turbine yaw misalignment by wake tracking" and by the same authorship of this paper which took place on the 12th of March 2014.

There we presented a proof of concept of a technical application of wake tracking different than the scope of this paper. During our presentation we mentioned for the first time that we have tested the double Gaussian but selected the single Gaussian for the purposes of wake tracking in the near wake. However no paper was published due to the preliminary character and due to unexpected delays in the permission process by the manufacturer of the wind turbine.

As you can see the date of presentation was prior to the publication date of Aitken's paper which is April 2014. Although the paper states that its final form was in October 2013, we have got acquainted of that paper in mid 2014. Furthermore, none of the authors of this paper took part in the review process of that paper, nor know of any previous publication of those ideas in a conference or early bird version of that paper.

With this we consider our approach as original as Aitken et al. is. In order to clarify this we propose to add a short reference to the context of the processing of the data and the oral contribution at the EWEA conference and hope that with this answer and its publication we accomplish with proper acknowledgement of the originality of these ideas by each author.

**3.10 Line 85 (and elsewhere)**

Why is the name of the wind farm enclosed in brackets and lowercase? Why is not written as "Alpha Ventus" wind farm?

**Answer**

Typically the wind farm name has been written in lowercase, therefore we decided to write it in the way we did.

**3.11 Line 125**

Figure 4 is somewhat misleading since it does not seem to be drawn to scale. The text states that the tower is located 8D or about 900 m from the turbine, but Figure 4 makes it look like the tower is less than 500 m from the turbine. I suggest redrawing the figure to scale.

**Answer**

This will be checked.

**3.12 Line 144**

Figure 6 depicts measurements of wind speed and direction. The instruments used to take these observations have some kind of measurement uncertainty. The measurement uncertainty should be quantified in the text, and Figure 6 should be modified to include error bars showing this uncertainty.

**Answer**

A detailed description of the uncertainty of the wind sensors used is explained in Westerhellweg-2012. We do not have access at the moment to the detailed data. However, we could add in the text the overall values of uncertainty.

Furthermore, the plots are given as indicative of the global inflow conditions and we believe that adding error bars will over complicate them without giving significantly more insight into the operational conditions.

Therefore we propose to add uncertainties in the text and let plots as they are.

**3.13 Line 156-157**

The language here seems vague: "The results looked qualitatively more similar. . ." What results are being referred to here? More similar in comparison to what? Similarly, the phrase "improvement of success" is also vague. Improvement over what? And what does "success" mean? That the fitting algorithm converged to a sensible solution?

**Answer**

Thank you for pointing this out. We have been here maybe too succinct and therefore we will extend the explanation with the guidance of your questions.

**3.14 Line 160-161**

What does it mean for a snapshot to have been tracked "successfully"? Why is 70% chosen as a cutoff point? This number seems completely arbitrary—are the authors able to justify this cutoff somehow?

**Answer**

Successfully means here that the fitting process has converged with the given bounds for all parameters and the convergence criteria. The 70% value is an ad-hoc value which we have defined in order to deal with "unsuccessful" fits. We have identified three main sources of failed attempts namely, a highly complex wind field, numerical issues of convergence and inexistence of wake deficit. The last one is related to the operational status of the turbine which at the time of applying the tracking procedure was not available. The results of sampled tests suggested a rather low effect of the first two sources, therefore we assume that any large lost of tracked centres should be revealing a downtime of the turbine. In conclusion the 70% is expected to guarantee that in a ten minute period the turbine was under normal operation during that percentage of time.

**3.15 Line 167**

Is there any significance to 2.6 degrees? Was this close to the average yaw error (the difference between the wind direction measured by the tower and

the yaw angle of the turbine) during the experiment? If so, that would be a neat result to point out.

**Answer**

It would be tempting to say that such lidar measurement can give a very accurate value of yaw misalignment bias. In part, this is in the lines of what we suggested in our proof-of-concept of "measuring wind turbine yaw misalignment by wake tracking" at the EWEA 2014. However, as we explained in the discussion of this paper, there is lack of some information to claim that this value represents only the mean misalignment. To do that, a proper experiment has to be deviced to discard some of the additional errors which could be contributing to the error of 2.6° found.

**3.16   Line 185-187**

The wake deviation delay seems analogous to the fact that the maximum velocity deficit is attained 1-2D behind the turbine, as noted in: J.F. Ainslie. Calculating the field in the wake of wind turbines. Journal of Wind Engineering and Industrial Aerodynamics, 27:213–224, 1988. and in Sanderse's literature review on the aerodynamics of wind turbine wakes. It would be interesting to point out the similarity between the deviation delay and the velocity deficit since both cases show that, in the real world, the impulse delivered by the rotor on the flow cannot occur instantaneously.

**Answer**

This is an interesting observation. The delay effect seems to be showing a fundamental difference between wind tunnel and real scale turbines. We have been thinking on some hypothesis explaining this difference, however we avoided to perform any speculation in the paper. An important point is how reproducable is this effect on other turbines. Such exercise could be performed by evaluating detailed CFD simulations and/or other measurement campaigns.

**3.17   Typos**

- Line 88: Change "minutes" to "minute".

- Line 108: Change "which" to "whose".

- Line 195: Change "analog" to "analogous".

**Answer**

These will be corrected. Thank you for taking the time to report this.

---

## Author Comment (AC2) · 28 Feb 2016

**Introduction**

Dear Reviewer,

thank you very much for your comments and suggestions for improvement of this paper to which you will find answers below. Additionally you will find our proposals to take them into account in a revised version of the paper which is to be prepared after consideration of the comments of all reviewers and the editor.

Sincerely,

Juan José Trujillo and Co-authors

**Major comments**

[Figure]

**Effects of thermal stability**

The selected site is well documented and the measurement period is long enough to ensure statistically converged results. One could regret that no classification of data versus the thermal stability was performed. Would it be possible to do this classification and observe the consequences on the results? If not, please at least comment on it within the document. What is the average turbulence intensity at hub height?

*Answer*

As you point it out stability effects could be of interest for this research. Indeed at processing time we have tried to evaluate the stability at the site, however we have come to the conclusion, that the setup and the accuracy of the available data (ten minutes averages of wind anemometer and termometers ranging from 30m to 100m) was not high enough to perform a stability analysis as needed for this research. Therefore, we think that it is not sensible to perform a stability classification due to large uncertainties in its estimation. We would propose to add some explanation (see details below) in the paper about this. Finally, we will comment on the turbulence intensity. We thank you for asking about this topic which we also think should be taken into account for the preparation of future research. An early consideration of this would encourage getting the proper data for performing a more accurate analysis of the stability effects.

*Background*

During our analysis we performed a rough estimation of the Monin-Obukhov length ($L_{MO}$). As explained in the paper, the horizontal wind speed measured at different heights was fitted to Businger-Dyer model of the vertical profile including thermal effects for each ten minutes data sets. In this process the three parameters of the model, namely, $L_{MO}$, friction velocity and surface roughness, were estimated by means of a least squares procedure. In this way we obtained a model for wind speed extrapolation above the highest measurement height of the meteorological mast for every ten minutes data set. The obtained results for the $L_{MO}$ showed non consistent results with

large variations and in some cases contradictory stability classification of consecutive data sets. Although this result could be due to the short averaging period of ten minutes, we interpreted this as an indication that, with the available data, the estimated stability would have a high uncertainty and therefore not contribute to an analysis with respect to stability.

Two sources support our findings about the high stability uncertainty and in some respect support our decision of not taking into account atmospheric stability in our analysis. First, Cañadillas et al. 2011 have shown that at FINO1 it is possible to obtain stability data with highly resolved sonic anemometer data with the more elaborated covariance method. Nevertheless, they found a high scatter of the results when comparing against a bulk Richardson approach including mean meteorological measurements of 30 minutes and sea surface temperature data. Second, in a personal communication with Gerald Steinfeld, Jens Tambke and Michael Schmidt, colleagues at the Energy Meteorology research group at ForWind - University of Oldenburg, who have experience with stability analysis at the FINO1 platform, they also expect a large uncertainty in the estimated stability. Moreover, they also coincide with Cañadillas that one of the main sources of uncertainty is the limitation in the setup where the lowest measurement height is too high (above 30m).

**References**

- Cañadillas, B.; Muñoz-Esparza, D. and Neumann, T., Fluxes Estimation and the derivation of the atmospheric Stability at the offshore mast FINO1 EWEA Offshore, 2011

**Skew angle assessment**

§4.1. Knowing the thrust coefficient, it is possible to assess the expected velocity deficit and so, the expected axial induction factor. One can then compare this velocity deficit with the one obtained in the present data base. One can also assess the expected

skew angle of the wake, which depends on the axial induction factor and the yaw angle. According to a rapid calculation, the skew angle of the wake should be 1.16 times higher than the yaw angle. However, this skew angle is not taken into account in the present study: why?

**Answer**

You are right that there is an expectation of a skew angle. As we mention in the discussion it is not possible to resolve the very small skew angles which are in consideration with the measurement setup in this research. As you point out we do not explain in detail the background of this to be concise in the paper. A solution could be to add an explanation into a short annex to the paper.

From theoretical considerations the wake deviation $(\chi_o)$ can be approximated as $\chi_o \simeq (0.6a + 1) \cdot \gamma$ (see vortex model in Burton et al. 2011) what derives into a skewness equal to $\psi \simeq 0.6a\gamma$. This behaviour is depicted in Fig. 1, where an approximate value of thrust coefficient $C_T$ is given for the case of no misalignment. This is done taking into account that based on the vortex model the thrust coefficient is rather unaffected by yaw errors in the order of magnitude shown in Fig. 1.

The shaded area depicts approximately the yaw misalignment values which are covered during this experiment. As a consequence the highest values of skewness that could be expected are below $\psi \simeq 3°$. However, this value lies already in the order of the typical uncertainty assumed for nacelle wind vanes. Therefore we conclude that the wake skewness can not be resolved from pure yaw misalignment measurement by the standard wind vane at the nacelle in this experiment.

In regards to the lidar measurements it is not possible to assess an uncertainty, as explained in the paper, since the lidar mounting error in azimuth cannot be exactly determined. Consequently, the orientation errors cannot be diminished to isolate the systematic effect of skewness. One could be tempted to make some interpretation of the values of $\Delta\gamma$ in Tables 2 and 3 in the paper, however this would be just speculation. In conclusion, the accuracy of the experiment is not enough to resolve the wake skewness for each misalignment angle.

This is an issue which could be approached in future campaigns with the possibility to measure well defined hard targets. In this way an accurate assessment of the mounting azimuthal error of the lidar system could be performed.

**Reference**

- Burton, T.; Sharpe, D.; Jenkins, N. & Bossanyi, E. & Sons, J. W. (Ed.) Wind Energy Handbook John Wiley & Sons, 2001

**Interpretation of very near position** Figure 7: the data scatter at the first position cannot be due to a physical behaviour of the near-wake. It is unlikely that the wake changes its position in such a magnitude really close to the rotor. One might conclude that the tracking method based on a Gaussian distribution is not appropriate to capture the very near wake and one could suggest not to interpret results from this very near position

**Answer**

Your observation is correct in that we should not draw any interpretation out of those data, specially with respect to the variance of the position. We tried to be careful in not deriving any interpretation or conclusion from the results of position variation at that location although they are presented for completeness. Nevertheless, in Sec. 5.1.2 we do an interpretation based on the mean position at that location due to the strong consistency of the trend of all mean values to lie on a straight line. We will revise the paper for any over-interpretation based on the data at this location.

In our discussion we do mention though, that we do not have enough information to discard the Gaussian fit as an appropriate method. Mainly, one issue of the measurement method from the nacelle is the strong reduction of the interrogated area closer to

the rotor. This reduces effectively the chances of success of the fitting procedure.

**Systematic shift in vane measurements** $4.2. the systematic shift angle of $3°$ between the wind vane and the wake path sounds as a systematic bias in the vane measurement. This is confirmed in $5.1.2 since a bias of $3°$ is mentioned. Why not taking this bias into account from the begin- ning of the data processing in order to remove this systematic shift?

*Answer* Your appreciation is correct and at some point we performed such "correction" in our analysis. However, as we mention in the introductory part of Sec. 5.1.2, it is not fully accurate to claim that the wind vane is the only source of this error. Our lidar could also have some bias in its azimuthal orientation which is impossible to verify in our setup. Therefore the only choice we have is to not correct the obtained wake centre positions, but clarify the different sources of error, as we did in our discussion.

**Shift/Delay** $5.1.1. "Delay" sounds like a temporal characteristic whereas the authors describe a spatial behaviour. The present study does not deal with unsteady properties of the near-wake since the data are time-averaged. "Shift" might be more appropriate in the current situation. This raises also the question of the settling time of the near-wake position after a modification of the yaw angle. In the present study, the history of the misalignment is not taken into account.

*Answer*

As you suggest, the word "Shift" seems more appropriate in the light of the steady character of the flow which is observed here.

In relation to settling time, you are right that a more accurate analysis should take either into account only situations with no yawing at all or take careful care of the manoeuvres. However, in our case we assume that this activity is not affecting the results due to the following reasons:

- Yaw manoeuvres under normal operation are typically of very small angular values.

- Assuming bulk wake advection speeds ranging from 3m/s to 10m/s the time for covering the 1.4D distance would range roughly from 60s to 16s. We expect that these values are well below the average periods between manoeuvres.

**Minor comments**

**Lidar probe volume** Please indicate the measurement volume (or length?) of the Lidar system : the LIDAR system spatially integrates along a line-of-sight distance, meaning that when you state to measure at a certain position, it is in reality an space average of along a given distance.

**Answer**

The measurements were performed with a pulse of 200ns and a range gate width of 100ns. With this setup the manufacturer estimates a probe length of approximately 34m. We will add a table with specifications of the lidar setup.

**Typos** Line 250 : "Vortices "instead of "vortexes"

**Answer**

It seems that "vortices" are winning the race against "vortexes" in google. Thanks for the tip.

**Fig. 1.** Approximated theoretical expectation of the effect of wind turbine misalignment on wake skewness with respect to induction (a) or thrust coefficient (Ct)